# Metabolic and Low-Grade Inflammation Risk in Young Adults with a History of Extrauterine Growth Restriction

**DOI:** 10.3390/nu16111608

**Published:** 2024-05-24

**Authors:** Laura Palomino-Fernández, Belén Pastor-Villaescusa, Inmaculada Velasco, María de la Cruz Rico, Juan Roa, Ángel Gil, Mercedes Gil-Campos

**Affiliations:** 1Metabolism and Investigation Unit, Reina Sofia University Hospital, Maimonides Institute of Biomedicine Research of Cordoba (IMIBIC), University of Cordoba, 14004 Cordoba, Spain; laurapafer95@gmail.com (L.P.-F.); belen.pastor@imibic.org (B.P.-V.); mercedes_gil_campos@yahoo.es (M.G.-C.); 2Primary Care Interventions to Prevent Maternal and Child Chronic Diseases of Perinatal and Developmental Origin (RICORS), RD21/0012/0008, Instituto de Salud Carlos III (ISCIII), 28029 Madrid, Spain; 3Department of Cell Biology, Physiology and Immunology, Reina Sofia University Hospital, Maimonides Institute of Biomedicine Research of Cordoba (IMIBIC), University of Cordoba, 14004 Cordoba, Spain; roarivas@gmail.com; 4Department of Biochemistry and Molecular Biology II, School of Pharmacy, “José Mataix Verdú” Institute of Nutrition and Food Technology (INYTA), University of Granada, 18016 Granada, Spain; mdcrico@ugr.es (M.d.l.C.R.); agil@ugr.es (Á.G.); 5Center of Biomedical Research, Instituto de Investigación Biosanitaria (IBS.Granada), University of Granada, 18016 Granada, Spain; 6Consorcio CIBER, M.P. Fisiopatología de la Obesidad y Nutrición (CIBEROBN), Instituto de Salud Carlos III (ISCIII), 28029 Madrid, Spain

**Keywords:** extrauterine growth restriction, inflammation, adipokines, insulin resistance, metabolism, nutrition

## Abstract

Children with a history of extrauterine growth restriction (EUGR), later at prepubertal age, exhibit an increased metabolic risk including risen insulin resistance and low-grade inflammation. However, the progression of such metabolic changes after puberty and the lasting health implications have not yet been investigated. The objective of this study was to ascertain whether young adults with a history of EUGR faced increased vulnerability to metabolic disorders. A study was conducted comparing a group of adults with a history of EUGR with a healthy reference group. A total of 110 young adults (36 from the EUGR group and 74 from the control group) were included. Anthropometric variables, blood pressure (BP), general biochemical parameters, plasma inflammatory biomarkers, and adipokines were assessed. Compared to the reference group, the EUGR group had a shorter height and body weight with higher lean mass and waist circumference, as well as a greater percentage of individuals with high BP. In addition, EUGR patients had higher values of insulin, HOMA-IR, nerve growth factor, and leptin, and lower levels of adiponectin and resistin. The present study suggests that young adults with a history of EUGR present increased metabolic risk factors therefore, clinical follow-up should be considered.

## 1. Introduction

Preterm birth, defined as a gestational age (GA) at birth of less than 37 weeks, is a major health problem associated with perinatal morbidity and mortality [1,2] involving 10% of deliveries [1,2]. This morbidity and mortality increase with decreasing GA and birth weight, as in the case of fetal growth restriction (FGR) or intrauterine growth restriction (IUGR) [3]. Both birth weight and GA are perinatal factors that have been extensively studied related to comorbidities and mortality. Birth weight is as an independent factor in mortality related to perinatal adverse events, lower postnatal growth, and alterations in neurological development, and a greater risk of future metabolic and cardiovascular diseases [4,5,6,7]. However, there is controversy among various authors concerning the significance of post-birth growth for the future development of preterm infants, as well as the appropriate methods for its assessment [8,9,10,11]. The most widely used definition to refer to the extrauterine growth restriction (EUGR) is having a weight below the 3rd percentile (p3) or p10 at 36 weeks corrected by GA and/or home discharge [12,13,14,15,16]. This is a frequent condition in neonatal units, with a prevalence of up to 60% depending on the cutoff point [9,15,16]. Although neonatal advances have significantly improved the survival of preterm infants, they are still insufficient to improve their growth rate and avoid important comorbidities [15,16].

Altered nutrition and growth during critical periods of life, such as in the third trimester of pregnancy, can affect the development of adipose tissue as well as the level of inflammation. This may confer an increased risk of future metabolic complications. One of the proposed mechanisms involves altered regulation of intrauterine adipogenesis, which is related to fetal stress and results in morphological and functional changes in adipose tissue and body composition [17]. These changes during the early adipogenesis are associated with an increased risk of metabolic pathology in later stages of life, such as high blood pressure (HBP) or insulin resistance [18,19]. This hypothesis would also be applicable to individuals with EUGR. Adipose tissue development and antioxidant mechanisms may be compromised in preterm infants with early postnatal undernutrition Therefore, strategies to modify the metabolism of adipose tissue during this window of opportunity could prevent the consequences in adulthood. The impact of EUGR on inflammatory status and oxidative stress is being studied as a determining factor in the development of metabolic diseases later in life [20]. Indeed, some adipokine changes and low-grade inflammation have been described in children at the prepubertal age (value of 1 on the Tanner scale) with a history of EUGR compared to their term counterparts [21,22].

Recent studies, some of which were conducted by our research group, have shown that prepubertal children with a history of EUGR present a greater risk of metabolic diseases than their preterm and term counterparts with no EUGR, showing higher blood pressure (BP) levels, proinflammatory biomarkers and deficiency in the antioxidant enzyme system [23,24,25,26,27,28,29]. In this way, the direct consequences of prematurity are differentiated from the consequences of the EUGR condition.

Therefore, the objective of the current research was to explore the metabolic and inflammatory status among young adults with a background of EUGR, which could be related to the development of metabolic disease in the future, compared to healthy youths without history of risk of disease.

## 2. Materials and Methods

### 2.1. Study Design

The present study was designed as a descriptive, analytical, cross-sectional study. Young adults were divided into two groups. The EUGR group included young adults with a Tanner pubertal stage ≥III, with a current signed consent form and without any metabolic disease that could influence the study results. This group include individuals from BIORICA-10 study, with 10-year follow-up of preterm newborns considered with EUGR according to the literature (with birth weight >p10 and <p3 at 36 weeks GA and/or at discharge from the neonatal unit) [12,16]. Newborns whose pathology was not associated with this neonatal condition were excluded.

The reference group included healthy young adult volunteers who met the following inclusion criteria at the time of measurement: pubertal stage (Tanner ≥ III regardless the age), born at term with adequate weight and height for GA (38–42 weeks GA and 2500–3500 g at birth), no relevant antecedents of metabolic disease, and confirmed to be free of any diseases through comprehensive physical and biochemical evaluation. The exclusion criteria for both groups were chronic disease and smoking.

This study was conducted in accordance with the Declaration of Helsinki and was approved by the Institutional Hospital Ethical Committee (human ethics approval number 3729_2020). The selected subjects were informed, and written consent was obtained. Participants under 18 years of age consented verbally, and written consent was signed by their parents or legal guardians.

### 2.2. Clinical History, Physical Examination, and Blood Pressure Measurements

Perinatal clinical data and personal and family health records were collected from the clinical history of both groups, as published previously [21,22]. The median GA for EUGR participants at birth was 29.5 weeks (interquartile range (IQR) 25–32). An extended physical exploration was conducted for all participants by the research physician. Pubertal status (Tanner III–V) was established through physical exploration and confirmed by sex hormone analyses. Body weight (kg), height (cm) and waist circumference (WC) (cm) were measured according to standardized procedures, with the subjects in minimal clothing, barefoot and having fasted overnight. Body mass index (BMI, kg/m^2^) and BMI z score were calculated for all participants using the standard growth percentile charts for the Spanish population [30]. Overweight and obesity were defined as BMI z scores >1 and >2, respectively, in participants under 18 years old according to the World Health Organization (WHO) [31] and BMI ≥ 25 and 30 kg/m^2^, respectively, in those older than 18 years. Underweight (UW) was defined as weight below the 3rd or 5th percentile, depending on the reference curves used, or weight below 80% of ideal weight for height or z score <−2 also considering WHO cut offs [32]. Body composition (fat mass, lean mass, and total body water) was measured by bioimpedanciometry (BC-418MA Body Composition Analyzer, TANITA^TM^, Tokyo, Japan). Fat mass index (FMI) was calculated as fat mass (kg) divided by height squared (m^2^).

Systolic blood pressure (SBP) and diastolic blood pressure (DBP) were measured twice by the same observer with a Dinamap V-100. The subjects rested in a supine position for ≥5 min, and a correct cuff was placed around the left arm. For participants under 18 years old, BP percentiles were established according to age and sex; HBP was defined as a BP >95th percentile (p95) and preHBP was defined as p90–95. In participants older than 18 years, HBP was defined as values ≥140/90 mmHg and preHBP between 120/80 and 139/89 according to the European Society of Cardiology/European Society of Hypertension Guide from 2018 [33].

### 2.3. Biochemical Parameters, Proinflammatory Biomarkers and Adipokines

Blood samples were collected from both the EUGR and reference groups following a 12 h overnight fasting period while the subjects were in a resting position lying down and using an indwelling venous line. One blood sample was obtained to analyze the general biochemical parameters within 2 h of collection, while the other sample was subdivided into aliquots and stored at −80 °C until the analysis of inflammatory markers and adipokines.

The general biochemical parameters included plasma total cholesterol (TC), high-density lipoprotein cholesterol (HDLc), low-density lipoprotein cholesterol (LDLc), apolipoprotein A and B, triacylglycerols (TG), iron, thyroid profile, and sex hormones. The carbohydrate metabolism markers used were glucose and insulin. Insulin resistance was calculated by the Homeostatic Model Assessment for Insulin Resistance index (HOMA-IR) calculated as insulin (mU/L) × glucose (mmol/L)/22.5. Pathological values are not well established in the literature. However, several researchers have concluded that values exceeding 3 indicate insulin resistance [34,35,36]. Analyses were performed by standardized laboratory methods using Architect c16000 and i2000SR autoanalyzers (Abbott Diagnostics^®^, Abbot Laboratories, Madrid, Spain). External and internal quality controls were performed according to hospital protocols. 

Inflammatory biomarkers were measured in plasma. C-reactive protein (CRP) levels were quantified using the autoanalyzer Architect c16000 (Abbott Diagnostics^®^, Abbott Laboratories, Madrid, Spain) by turbidimetric immunoassay with latex particles. Interleukin (IL)-6, IL-8, IL-1β, monocyte chemotactic protein type 1 (MCP-1), neural growth factor (NGF), tumor necrosis factor α (TNF-α), plasminogen activator inhibitor type 1 (PAI-1) and adipokines such as leptin, resistin, and adiponectin were analyzed in duplicate on a Luminex 200 system with XMap technology (Luminex Corporation, Austin, TX, USA) using human monoclonal antibodies (Milliplex Map Kit, Millipore, Billerica, MA, USA). Assay reproducibility was tracked to ensure that the values obtained were reliable (inter- and intra-assay coefficient of variation < 10%).

### 2.4. Statistical Analysis

The sample size was derived from the BIORICA study conducted in prepubertal stage by our research group [21,22]. It was calculated assuming a 30% difference in the mean of the main outcome variables (such as insulin) between control and EUGR groups, with an alpha error of 0.05 and a statistical power of 0.90.

Data are reported as counts and proportions for categorical variables and as the mean ± standard error (SE) for continuous variables. All continuous variables were tested for normality and outliers using Q-Q plots and histograms. One value for HOMA-IR identified as an outlier (>Q3 + 1.5 × IQR) was removed. Proportions were compared with the χ2 test. For comparisons of quantitative data between groups, analysis of covariance (ANCOVA) adjusted by age, sex, and FMI (as appropriate) was performed. The IL-8 values were log10 transformed to mitigate substantial dispersion in the data. However, the data are presented as untransformed values to ensure a clear understanding. The normality of residuals for ANCOVA models was assessed using histograms. The mean values adjusted from the models were extracted. To assess the differences by age between groups, the Mann-Whitney U test was used; therefore, this variable is expressed as the median and IQR.

To identify the metabolic and inflammatory variables strongly associated with the EUGR, binary logistic regression models (healthy group as reference value = 0) were used to estimated odds ratio (OR) and 95% confidence intervals (95% CIs). First, all variables that were significantly different between groups with *p* ≤ 0.02 were individually assessed using logistic regression analysis. Second, using the backward method selection, the quantitative variables with a *p* value ≥ 0.15 were entered simultaneously in the model using the Wald statistic to eliminate one by one from the model until the adjusted ORs of the variables with the strongest associations were obtained.

Data analysis was carried out using SPSS Statistics 25 software (IBM SPSS, Inc., IBM, New York, NY, USA). A *p* value < 0.05 indicated statistical significance.

## 3. Results

Thirty-seven EUGR participants were initially recruited. However, one female was excluded because she was still in a prepubertal stage. Seventy-six participants were initially recruited for the reference group. However, one participant was excluded due to ongoing growth hormone treatment, and another participant was excluded based on a diagnosis of polycystic ovarian syndrome. Consequently, the final study sample comprised 110 participants (36 EUGR, 74 controls). No participants withdrew from the study. A flow chart diagram is included in Figure 1.

The demographic, anthropometric, body composition and BP data are presented in Table 1. Compared with the reference group, the EUGR group showed decreased height, body weight, and lean mass, coupled with an elevated WC. Moreover, the EUGR group also had a greater percentage of UW adults in contrast to the controls. Relative to BP, the EUGR subjects displayed higher levels of SBP, and the proportion of individuals with HBP was greater in comparison with the reference group (16.7% vs. 1.4%, *p* = 0.007) (Figure 2).

Regarding the lipid profile, a significant decrease in TC was found in the EUGR group. Additionally, the EUGR group exhibited higher values of insulin and HOMA-IR than did the reference group (Table 2). Importantly, the prevalence of insulin resistance (HOMA-IR > 3) was higher in EUGR participants than in controls (22.2% vs. 2.7%, *p* = 0.001).

Notably, lower levels of TNF-α and IL-8, as well as higher NGF plasma concentrations were observed in the EUGR group compared with the reference group. However, no differences were found in other inflammatory parameters, such as CRP, PAI-1, or MCP-1 (Table 2). For IL-6 and IL-1β, more than 50% of the values were out of the limit of sensitivity (0.2 pg/mL and 0.4 pg/mL, respectively), rendering these biomarkers inconclusive and thus excluded from analysis. Concerning adipokines, the EUGR group showed lower levels of adiponectin and resistin but exhibited higher levels of leptin than did the control group (Table 2).

The data obtained by logistic regression analysis are shown in Table 3. An association between a history of EUGR and the risk of elevated SBP values and higher WC was observed. In addition, the EUGR group presented a greater risk of decreased weight and height in adulthood. The parameters with the strongest associations with the EUGR were WC (OR: 1.824, 95% CI 1.3–2.5), body weight (OR: 0.620, 95% CI 0.48–0.80), and NGF levels (OR: 4.092, 95% CI 1.50–11.16) according to the backward method. Similarly, the UW, HBP, and insulin resistance conditions were strongly associated with the EUGR group. Moreover, subjects with a history of EUGR were 30 times more likely to have UW (OR: 30, 95% CI 3.6–250, *p* = 0.002), 16 times more likely to have HBP (OR: 15.9, 95% CI 1.8–140.5, *p* = 0.013), and 10 times more likely to have insulin resistance (OR: 10, 95% CI 1.99–50.04, *p* = 0.005).

## 4. Discussion

The present study revealed that young adults with a history of EUGR exhibit reduced weight and height in adulthood, with a higher prevalence of HBP and greater insulin resistance risk than healthy controls.

First, it is noteworthy that upon culmination of the growth period, the group with a history of EUGR had a substantial percentage of UW patients and, in general, lower height and weight in comparison to the reference group, regardless of age and sex. Therefore, it seems that the EUGR group maintains a low weight status risk that persists into adulthood. Underweight and underheight have been previously described to be correlated with prematurity. Finken et al. (2006) reported that approximately 20% of a total of 380 adolescents with a history of prematurity of adequate weight for GA developed postnatal growth retardation and a greater prevalence of short height [37]. Similarly, 10–15% of adults with a history of IUGR continue to have a short stature in adulthood, and growth hormone treatment has even been approved for this condition [38,39]. Moreover, it has been observed that prepubertal EUGR children present a greater risk of having lower weight, which has historically been associated with prematurity rather than specifically with EUGR or IUGR itself [28]. In the present study, almost 30% of the participants in the EUGR group were UW in adulthood.

Furthermore, despite a lower body weight and a lower percentage of fat mass, the EUGR group had a significantly greater WC. This phenomenon could be linked to an abnormal distribution of fat, which is also observed in individuals with obesity and contributes to cardiometabolic risk. The accumulation of visceral adipose tissue results in increased infiltration of immune cells and the release of vasoconstrictor mediators, thereby contributing critically to the development of endothelial dysfunction, HBP, and vascular stiffness. Therefore, fat distribution rather than total body weight is a key determinant of CVD risk [40,41,42].

Relatedly, one of the most relevant findings of the present study was the high levels of SBP observed in the participants with a history of EUGR compared with the reference group. Almost 17% of participants showed HBP in the EUGR group versus a single person in the reference group. Similar results have already been described in prepubertal children with a history of EUGR [28], and HBP has been extensively documented in pediatric patients with IUGR. This condition seems to persist into adulthood and as mentioned, shares common pathophysiological mechanisms such as hemodynamic redistribution and cardiovascular remodeling in response to inadequate nutritional adaptation [43,44]. Individuals with IUGR are also more likely to present metabolic changes that have been associated with the subsequent development of HBP, insulin resistance, diabetes mellitus, and hyperlipidemia, which are often associated with abdominal obesity, although the related mechanisms are still unknown [40,45]. Thus, as in IUGR patients, the EUGR group during adulthood seems to have a greater risk of developing HBP than does general population.

Regarding the biochemical profile, in the present study, the EUGR group did not exhibit any significant difference in the lipid profile. Lower HDLc levels have been previously described in the prepubertal EUGR population than in children with a history of prematurity without EUGR and in a healthy group [22,28]. Some studies including large cohorts suggest that childhood height growth and adipose tissue may have more relevance than prenatal growth to the blood lipid profile [46,47,48,49]. However, further studies are needed to better understand the role of fetal growth in the changes in lipid metabolism.

In relation to the glycemic profile, insulin and HOMA-IR were significantly greater in the EUGR group than in the controls without pathological results. In the prepubertal EUGR populations from Ortiz’ and Ordoñez’s studies [22,28], they described higher levels of glucose than in the preterm and control groups but still without alterations in insulin values. It appears that these differences become more pronounced after puberty, as evidenced by the presence of insulin resistance in 22% of the EUGR group in the present work. These findings, in conjunction with those of other studies, provide further evidence that early malnutrition during extrauterine life, irrespective of birth weight, has a negative effect on insulin metabolism and glucose tolerance throughout an individual’s lifespan, even within the context of a normal BMI range [22,28,50]. Therefore, it is advisable to prevent obesity and other cardiometabolic comorbidities in individuals exposed to early undernutrition. It has previously been shown that in pediatric populations investigated for cardiovascular risk (elevated BP, excess weight or altered lipid profile), the HOMA-IR index is one of the most important factors associated with WC and SBP or other cardiovascular risk factors [51], even when body weight is not high enough to affect other obesity-related risk markers [52,53]. Accordingly, a correlation between abdominal perimeter and HOMA-IR was found in this study. In good agreement, a study conducted with 75,000 Brazilian adolescents revealed that high abdominal circumference is closely related to insulin resistance, with an increase in abdominal circumference predicting higher levels of IR [53].

Adiponectin is one of the most extensively studied adipokines, primarily due to its inverse association with the risk of metabolic syndrome (MetS) development [52,54,55]. It has anti-atherosclerotic, antithrombotic, antioxidant, and anti-inflammatory properties [52,56]. Hence, insulin resistance, HBP, diabetes, or the development of CVD have been associated with lower plasma adiponectin levels regardless of body weight [54,55,57]. Additionally, low adiponectin levels and high visceral adiposity might affect insulin resistance and β-cell dysfunction [58]. Previous studies in prepubertal children with a history of EUGR showed lower adiponectin levels and higher resistin concentrations in plasma compared with a healthy group of children without differences in other adipokines, such as leptin [27]. In the present work, we observed that the EUGR group had a different adipokine profile than the healthy group, characterized by lower levels of adiponectin and resistin and higher levels of leptin. Low birth weight has previously been linked with high leptin and low adiponectin levels [59]. However, some studies have reported inconsistent findings regarding resistin in relation to low perinatal weight and obesity or overweight [59,60,61,62]. Therefore, the role of this adipokine in regulating insulin sensitivity and adipogenesis during the perinatal period is not yet fully understood [63]. Another hypothesis is that resistin acts as an insulin antagonist, and thus, insulin resistance might suppress resistin levels [60], but further research is necessary to elucidate the precise role of resistin in these metabolic alterations. 

In terms of the inflammatory profile, certain outcomes documented within this cohort of young adults present challenges in terms of interpretation, especially in circumstances where an anticipation of low-grade inflammation exists, as observed in MetS patients. Higher levels of NGF and lower levels of TNF-α and IL-8 were detected in the EUGR group than in the reference group. This observation may be explained by the mechanism by which binding NGF to TLR-activated monocytes leads to a reduction in the production of inflammatory cytokines and increases the production of anti-inflammatory cytokines as a mechanism to reduce chronic inflammation [64]. Most of these inflammatory parameters and adipokines are produced in adipose tissue, and they can interfere with multiple inflammation mechanisms, innate immunity, and a broad spectrum of antimicrobial activity modulating immunity and limiting inflammation induced by microbes [65]. Hence, these differences could be explained by a deregulation of inflammation characterized by an imbalance between pro- and anti-inflammatory factors, consequently altering the first line of immunity.

The present study has several strengths. Most of the data were available in all studied groups, thereby enhancing the potential of this study to contribute to our understanding of the metabolic status among young adults with EUGR conditions during the perinatal period. This sample was exhaustively selected following all established criteria to classify individuals with EUGR. Likewise, we ensured that all participants had completed puberty and did not have any other disease that could affect the study results. Also, our study has some limitations. Indeed, there is no consensus about how the postnatal growth of preterm infants should be monitored, and the optimal growth pattern has not yet been determined. However, different cutoff points have been defined in relation to development of some comorbidities [66,67,68]. Another point to consider is that the reference group were slightly older and with a greater proportion of females compared to the EUGR group. However, all of them were followed up after 10 years considering the previous range age as prepubertal, and now being at least in III Tanner stage of puberty according to the inclusion criteria. Maintaining a 10-year follow-up is difficult and especially within a group of preterm patients without EUGR and other comorbidities. Given that EUGR and reference groups exhibited the greatest differences during prepuberty stage [27,28,29], comparisons between these extreme groups were more appropriate, not considering a third and small group of preterm without EUGR.

## 5. Conclusions

In conclusion, the persistence of altered metabolic conditions in individuals with a history of EUGR throughout life may contribute to a greater risk of metabolic disorders and cardiovascular diseases in the future. To the best of our knowledge, this is the first study assessing metabolic and inflammatory status in young adults with a history of EUGR; therefore, it is necessary to advance this line of research to establish a medical follow-up to prevent diseases in this population.

## Figures and Tables

**Figure 1 nutrients-16-01608-f001:**
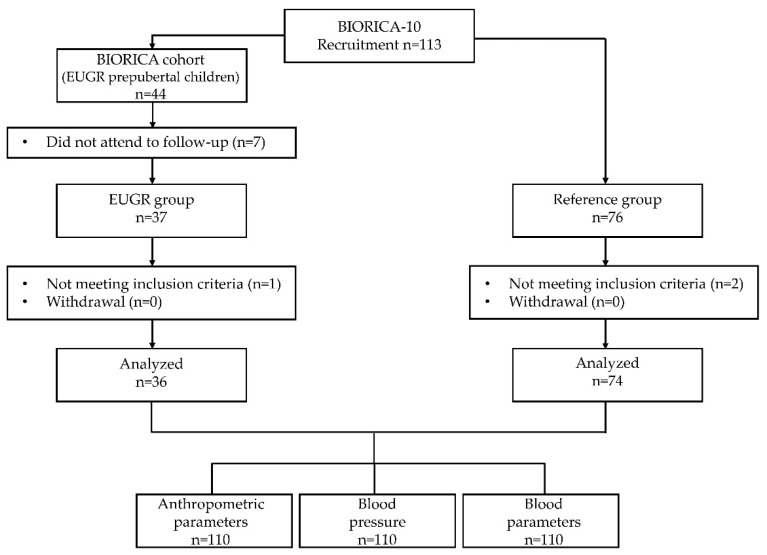
Flow chart diagram for all participants in BIORICA-10 study.

**Figure 2 nutrients-16-01608-f002:**
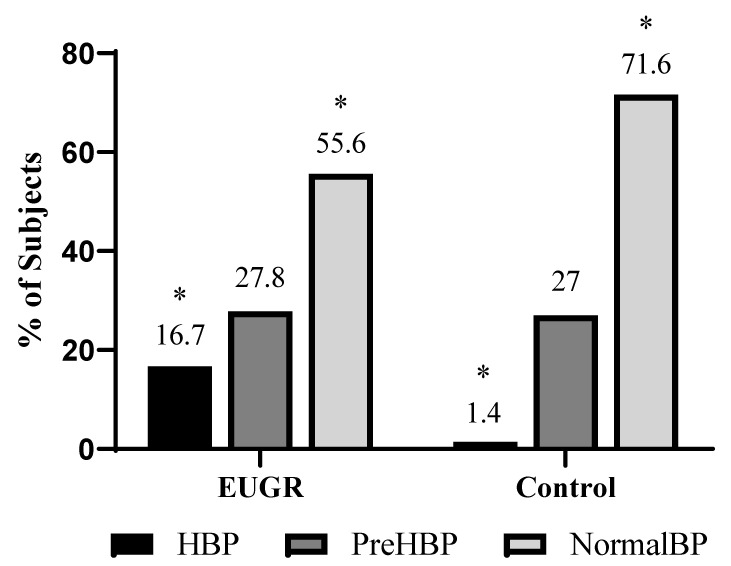
Percentages of young adults with a history of EUGR with normal levels of blood pressure (BP), pre high blood pressure (preHBP) and high blood pressure (HBP) compared with a healthy control group. * Differences were significant with a *p* value < 0.05 (χ2 test).

**Table 1 nutrients-16-01608-t001:** Demographic and anthropometric parameters and blood pressure in young adults with a history of extrauterine growth restriction (EUGR group) and a healthy group born at term (Control group).

General Variables	EUGR Group (n = 36)	Control Group (n = 74)	*p* Value
Sex (M/F)	25/11	35/39	**0.029** ^a^
Age (years)	17 (6)	21(2)	**<0.001** ^b^
Weight (kg)	53.4 ± 2.2	64.4 ± 2.1	**0.008** ^c^
Height (cm)	161.4 ± 1.5	168.7 ± 0.9	**<0.001** ^c^
BMI (kg/m^2^)	21.8 ± 0.4	22.4 ± 0.6	0.480 ^c^
NW/OW/UW (%)	52.8/19.4/27.8	78.1/20.5/1.4	**<0.001** ^a^
WC (cm)	76.4 ± 1.7	68.3 ± 1.1	**0.001** ^c^
Fat Mass (kg)	11.9 ± 1.1	14.2 ± 0.7	0.121 ^c^
Fat mass (%)	19.6 ± 1.2	22.7 ± 0.7	0.054 ^c^
FMI (kg/m^2^)	4.7 ± 0.4	5.0 ± 0.2	0.532 ^c^
Lean mass (kg)	45.2 ± 1.5	50.1 ± 0.9	**0.016** ^c^
Lean mass (%)	80.4 ± 1.2	77.3 ± 0.7	0.054 ^c^
SBP (mmHg)	122.2 ± 2.08	115.9 ± 1.2	**0.022** ^c^
DBP (mmHg)	67.6 ± 1.8	67.5 ± 1.07	0.968 ^c^
HR (lpm)	78.4 ± 3.5	70.8 ± 2.06	0.097 ^c^

Data are expressed as n (%) for categorical variables and adjusted mean ± SE or median (IQR) (age) for continuous variables. BMI, body mass index; DBP, diastolic blood pressure; FMI, fat mass index; HR, heart rate; NW: normal weight; OW: overweight; SBP, systolic blood pressure; UW: underweight; WC, waist circumference. Differences between the EUGR and control groups were assessed using the ^a^ χ2 test, ^b^ Mann-Whitney U test or ^c^ Analysis of covariance (ANCOVA) adjusting for age, sex, and FMI (when appropriate). Significant differences in bold (*p* < 0.05).

**Table 2 nutrients-16-01608-t002:** Plasma biomarkers concentrations related to glucose and lipid metabolism, adipokines and other inflammatory cytokines measured in a young adult group with a history of extrauterine growth restriction (EUGR), compared with a healthy control group.

	EUGR Group (n = 36)	Control Group (n = 74)	*p* Value ^a^
Glycemic metabolism and insulin sensitivity			
Glucose (mg/dL)	86.2 ± 1.7	85.2 ± 1.04	0.681
Insulin (µU/mL)	9.6 ± 0.7	7.2 ± 0.4	**0.012**
HOMA-IR	2.0 ± 0.2	1.5 ± 0.09	**0.025**
Lipid parameters			
Triacylglyerols (mg/dL)	59.6 ± 6.9	77.2 ± 4.1	0.053
Cholesterol (mg/dL)	150.3 ± 5.7	165.9 ± 3.4	**0.040**
HDLc (mg/dL)	57.2 ± 2.4	57.9 ± 1.5	0.827
LDLc (mg/dL)	81.07 ± 4.7	92.6 ± 2.8	0.064
Apo A (mg/dL)	128.2 ± 4.09	131.2 ± 2.4	0.576
Apo B (mg/dL)	58.3 ± 3.6	63.8 ± 2.1	0.240
Adipokines and inflammatory parameters			
Adiponectin (mg/L)	16.3 ± 2.6	25.7 ± 1.5	**0.006**
Resistin (μg/L)	12.6 ± 2.3	33.7 ± 1.4	**<0.001**
Leptin (μg/L)	4.6 ± 0.5	3.0 ± 0.3	**0.023**
PAI-1 (μg/L)	15.9 ± 1.9	10.8 ± 1.2	0.050
NGF (pg/mL)	3.2 ± 0.3	2.4 ± 0.2	**0.016**
MCP-1 (pg/mL)	94.06 ± 8.9	106.7 ± 5.4	0.281
TNF-α (pg/mL)	1.7 ± 0.2	2.5 ± 0.1	**0.002**
IL-8 (pg/mL) ^b^	0.5 ± 2.09	6.3 ± 1.3	**0.046**
CRP (mg/L)	0.6 ± 0.5	1.4 ± 0.3	0.239

Data are expressed as the adjusted mean ± SE. ^a^ Differences between the EUGR and control groups were assessed using analysis of covariance (ANCOVA). All ANCOVA models were adjusted for age, sex, and FMI. ^b^ IL-8 was log10 transformed for the analyses. Significant differences in bold (*p* < 0.05). Apo A, apolipoprotein A; Apo B, apolipoprotein B; CRP, C-reactive protein; HDLc, high-density lipoprotein cholesterol; HOMA-IR, homeostatic model assessment index for insulin resistance; IL-8, interleukin 8; LDLc, low-density lipoprotein cholesterol; MCP-1, monocyte chemotactic protein type 1; NGF, neural growth factor, PAI-1, plasminogen activator inhibitor type 1; TNF-α, tumor necrosis factor-α.

**Table 3 nutrients-16-01608-t003:** Analysis of association between the main metabolic and inflammatory markers and the condition of EUGR.

	OR	IC (95%)	*p* Value ^a^
Height	0.936	(0.9–0.98)	0.002
Weight	0.933	(0.9–0.97)	<0.001
WC	1.059	(1.02–1.11)	0.007
SBP	3.125	(1.11–8.8)	0.030
HOMA-IR	1.839	(1.19–2.84)	0.006
Adiponectin	0.871	(0.82–0.93)	<0.001
Resistin	0.754	(0.67–0.85)	<0.001
Leptin	1.144	(1.01–1.3)	0.037
NGF	2.041	(1.38–3.03)	<0.001
TNF-α	0.222	(0.09–0.49)	<0.001

Data are expressed as odds ratio (OR) values and 95% confidence intervals (CI). ^a^ Binary logistic regressions performed on those variables with clinical differences between EUGR and control healthy group (*p* ≤ 0.02). The quantitative variables were entered simultaneously in the model using the Wald statistic and those with the strongest associations were obtained. HOMA-IR, homeostatic model assessment index for insulin resistance; NGF, neural growth factor; SBP, systolic blood pressure; TNF-α, tumor necrosis factor α; WC, waist circumference.

## Data Availability

Data are contained within the article.

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
