# Peer review of "Metabolic and Low-Grade Inflammation Risk in Young Adults with a History of Extrauterine Growth Restriction"

_nutrients, 2024, doi:10.3390/nu16111608_

Round 1

Reviewer 1 Report

Comments and Suggestions for Authors

Thank you for the opportunity to review this manuscript. I commend the authors on their long term follow up of this cohort and investigation their individual biomarkers for metabolic risk factors. However, it is a comparison between a subgroup of adolescents born preterm who did not grow well in neonatal care and a control group who were born at term. Therefore, we cannot conclude that differences at this age are due to EUGR and not preterm birth. 

There are several questions that should be further addressed and my suggestions for improvement follow:

Abstract

Page 1 Line 33 As this research is more about individual biomarkers for metabolic risk factors it might be more appropriate to say “risk factors” instead of “metabolic disease”

Background/Introduction

Page 1 Line 38. Reference 1. Is a systematic review of growth patterns and body composition in former extremely low birth weight (ELBW) neonates until adulthood. It does not provide good support for this statement.

Similarly, Reference 2. Is an overview of adult health outcomes after preterm birth published in 2020.
Please consider using a recent WHO reference for the % of children born preterm worldwide. (e.g., Born too soon: decade of action on preterm birth. World Health Organization, 2023)

Page 1 Line 42 “Both birth weight and gestational age are perinatal factors that have been extensively studied [4-7]”
Please clarify what birthweight and gestational age have been studied in relation to?

Page 1 Line 43-45 “However, there is controversy among various authors concerning the significance of post-birth weight for the future development of preterm infants, as well as the appropriate methods for it assessment [8-11].
Consider replacing the word “weight” with “growth”.

Page 2 Line 47 In 2020 there was a publication and call from a large international group of neonatal nutrition researchers
(your reference 9) to no longer use the term extrauterine growth restriction (EUGR). The reasons for this are well described in this paper so I wonder why the authors chose to use this definition rather than a measure of postnatal growth such as z-score change between two time points?

Page 2 Line 58 “…and results in morphological and functional changes in adipose tissue and body composition.” At what stage of life? Birth? Persistent?

Method

“EUGR according to the literature (with birth weight >p10 and and <p3 at 36 weeks gestational 84 age and/or at discharge from the neonatal unit)” [23, 30]. Reference 23 is about Metabolic changes in prepuberty children with extrauterine growth restriction but is in Spanish so I cannot read it. Reference 30 is about Impaired Antioxidant Defence Status Is Associated With Metabolic Inflammatory Risk Factors in Preterm Children With Extrauterine Growth Restriction also in Spanish. What growth chart was used to assess EUGR?

Please explain why BMI z scores ≥1 and ≥2 were used to assess Overweight and obesity but <3rd or 5th percentile was used or weight below 80% of ideal weight for height to assess Underweight? This seems inconsistent.

Does the Spanish growth chart go up to 21 years? Otherwise how was BMI z-score calculated for those over 20 years old?

Why not use a preterm non EUGR comparison rather than or as well as term. Why was this group from the original study left out of this follow up?

What is the statistical power of this sample size?

Results

The EUGR group had more males and was substantially younger meaning BMI categories etc was assessed using different methods. This is a major limitation of the study. In this case, adjusting for age and sex is not sufficient mitigation. Wt, length and HC different due to age difference

What was the age range? – large SD in the EUGR group of 6 meaning there may be 11 year olds. Is this correct?

Please add the gestational age at birth of the EUGR group?  

Discussion

Page 7 Line 268 State how many patients had HBP in the control group.

Page 7 Line 270 At what age was HBP observed in patients after IUGR?

Page 8 Line 317 “perinatal birth weight”? What does this mean – just birthweight perhaps?

Table

Table 1 and Table 2 - Mark the analyses that were adjusted for age, sex, and FMI

Table 3 – Please clarify – the odds of what in relation to height, weight, etc – low, high?

Figures

There should be a flow chart of recruits and those excluded.

The large number of abbreviations used in this manuscript reduce the readability.

Comments on the Quality of English Language

Minor improvements to the quality of the english needed.

Way too many acronyms

Reviewer 2 Report

Comments and Suggestions for Authors

The authors of the paper entitled "Metabolic and low-grade inflammation risk in young adults with a history of extrauterine growth restriction" attempted to demonstrate that young adults with a history of EUGR have a higher risk of metabolic disorders. It is an important and interesting study. Authors have several publication in this field. The article is well written and the results are correctly presented. The article expands knowledge about the impact of EUGR on the further health status of people who have had it in history. However, it is not very innovative.

I have to raise some concerns:

Major concerns:

·   - why is the size of the control and study groups so significantly different - does it distort the results? how were group sizes determined? what statistical test was used?

·   -why did patients in the control group differ in age from patients with EUGR? Couldn't you get a group more similar in age?

·  - there was no comparison between the sexes, and in my opinion this could significantly affect the results.

Minor concerns:

·Keywords: remove the numbers in front of individual words

·- References in the manuscript should be in square brackets- revise throughout the manuscript

·   - remove the double dot from subsections in the Materials and Methods section

Round 2

Reviewer 1 Report

Comments and Suggestions for Authors

The authors have addressed almost all comments and acknowledged the limitations. The study has serious limitations, but these have now been more  appropriately communicated.

Comments on the Quality of English Language

No further comments in addition to the first review